# Optimal Temporal Filtering of the Cosmic-Ray Neutron Signal to Reduce Soil Moisture Uncertainty

**DOI:** 10.3390/s22239143

**Published:** 2022-11-25

**Authors:** Patrick Davies, Roland Baatz, Heye Reemt Bogena, Emmanuel Quansah, Leonard Kofitse Amekudzi

**Affiliations:** 1Department of Meteorology and Climate Science, Kwame Nkrumah University of Science Technology, Kumasi AK-385-1973, Ghana; 2Leibniz Centre for Agricultural Landscape Research (ZALF), Eberswalder Str. 84, 15374 Müncheberg, Germany; 3Forschungszentrum Juelich GmbH, 52425 Juelich, Germany

**Keywords:** soil moisture, cosmic ray neutron sensor, synthetic neutron flux, Savitzky–Golay filter, Kalman filter, median filter

## Abstract

Cosmic ray neutron sensors (CRNS) are increasingly used to determine field-scale soil moisture (SM). Uncertainty of the CRNS-derived soil moisture strongly depends on the CRNS count rate subject to Poisson distribution. State-of-the-art CRNS signal processing averages neutron counts over many hours, thereby accounting for soil moisture temporal dynamics at the daily but not sub-daily time scale. This study demonstrates CRNS signal processing methods to improve the temporal accuracy of the signal in order to observe sub-daily changes in soil moisture and improve the signal-to-noise ratio overall. In particular, this study investigates the effectiveness of the Moving Average (MA), Median filter (MF), Savitzky–Golay (SG) filter, and Kalman filter (KF) to reduce neutron count error while ensuring that the temporal SM dynamics are as good as possible. The study uses synthetic data from four stations for measuring forest ecosystem–atmosphere relations in Africa (Gorigo) and Europe (SMEAR II (Station for Measuring Forest Ecosystem–Atmosphere Relations), Rollesbroich, and Conde) with different soil properties, land cover and climate. The results showed that smaller window sizes (12 h) for MA, MF and SG captured sharp changes closely. Longer window sizes were more beneficial in the case of moderate soil moisture variations during long time periods. For MA, MF and SG, optimal window sizes were identified and varied by count rate and climate, i.e., estimated temporal soil moisture dynamics by providing a compromise between monitoring sharp changes and reducing the effects of outliers. The optimal window for these filters and the Kalman filter always outperformed the standard procedure of simple 24-h averaging. The Kalman filter showed its highest robustness in uncertainty reduction at three different locations, and it maintained relevant sharp changes in the neutron counts without the need to identify the optimal window size. Importantly, standard corrections of CRNS before filtering improved soil moisture accuracy for all filters. We anticipate the improved signal-to-noise ratio to benefit CRNS applications such as detection of rain events at sub-daily resolution, provision of SM at the exact time of a satellite overpass, and irrigation applications.

## 1. Introduction

Soil moisture is a critical component of the climate system. It plays a crucial role in moisture distribution and energy through soil–atmosphere feedback [1]. Furthermore, soil moisture influences the evolution of the planetary boundary layer during the day [2], the surface Bowen ratio, convective available potential energy [3,4], and cloud formation [5]. It is vital for crop production in the agriculture sector when the right amount of water and conditions are known. The sustainable production of agricultural products is dependent on the effective management of agricultural water. The availability of water resources for agricultural production is significantly threatened by global climate change [6,7]. For improved use of this resource in agriculture, it is imperative to monitor soil moisture, which is a crucial variable in irrigation management and hydrological modeling, groundwater recharge, and flood and drought forecasting.

Several techniques for estimating point and spatial soil moisture have emerged over the past decades. The point or in situ source measurement methods include Time Domain reflectometry (TDR), Frequency Domain Reflectometry (FDR), and gravimetric soil sampling. A comprehensive review of in situ soil moisture estimation technique was conducted by [8], emphasizing their strengths and weakness. One common limitation of these methods is their uncertainty in spatial representativeness, as they only measures 1 dm${}^{3}$. The extent of coverage of these techniques in depth and area is limited due to the substantial human effort and destructive nature of the installation and high maintenance costs. In recent years, remote sensing satellites made substantial improvements in soil moisture observation [9]. However, although they have a good spatial coverage, they offer only shallow soil moisture (0 to 3 cm [10]) and a temporal resolution of 1.5–4 days [11]. These significant temporal and spatial resolution shortcomings make practical applications of retrieved hydrologic products difficult for users.

CRNS is a non-invasive technique for estimating soil moisture on relevant scales that have shown promising results [12,13]. Cosmic ray neutron sensors work using principles of nuclear physics. Fast neutrons express a high affinity to hydrogen due to the mass of a neutron being similar to that of a nucleus of the hydrogen atom [13]. Soil moisture can be estimated using the CRNS method by considering the inverse relationship between soil moisture and aboveground epithermal neutrons [14]. The CRNS estimates the average areal soil moisture over a radius of 120–240 m and a depth ranging between 15 and 80 cm depending on the moisture content of the soil and other parameters (aboveground biomass, atmospheric humidity, etc.). The introduction of the CRNS method for estimating soil moisture bridged the gap between point measurement and satellite estimation of soil moisture. Previous studies showed excellent agreement between CRNS-estimated soil moisture in comparison to co-located point-scale sensor networks covering a similar footprint at sites with advantageous conditions [15,16]. It has been shown that performance is lower at sites with less favorable conditions (locations with high atmospheric pressure, high biomass density, and humid climates) [17,18,19]. The external factors that influence the neutron count include atmospheric pressure [20], incoming neutron flux (see, e.g., [15,17]), specific humidity [21], and biomass [17,22]. Several factors contribute to the presence of neutrons in the atmosphere, including the attenuation of incoming neutrons from space by air molecules. As a result, surface neutrons change with changes in air mass. In addition, the atmospheric water content influences neutron abundance at the surface in two ways—integral water content in the whole air column moderates neutrons above the neutron detector. The presence of near-surface water vapor in the sensor’s footprint reduces the soil albedo component. Standard correction techniques are proposed for correcting these external or atmospheric feedbacks (see [15,17,21,22]). Other sources of uncertainty in neutron flux are calibration parameter (*N*${}_{0}$), detector size, and neutron count rate, which follow Poisson statistics [15]. Various neutron detectors exist of different sizes and efficiencies. Typically, a larger detector volume improves the counting statistics and thus reduces the uncertainty of the soil moisture product. A Poissonian distribution applies to a non-continuous quantity such as a neutron count, becoming Gaussian (i.e., normal) distribution for neutron counts above 30 counts per hour (cph) [23]. Conditions such as wet environment, dense vegetation and low elevation affect the CRNS detection of soil moisture within its accuracy limit [24]. In such environments, the error is mainly caused by the uncertainty of the neutron count and other hydrogen sources. However, since the count rate is inversely related to soil moisture, drier soils lead to more accurate measurements. Neutron counts are usually recorded as hourly totals, which are subsequently converted to soil moisture. Commercially available detectors for cosmic rays still exhibit a high degree of noise at hourly resolution, leading to high uncertainties in soil moisture measurements (example, [25]). This is usually resolved by applying a temporal filter to reduce the uncertainties in soil moisture.

Several studies have examined how some environmental factors that affect neutron intensity, as seen in the previous paragraph. These early studies focused on reducing uncertainties caused by environmental factors such as biomass, water vapor, and incoming neutron intensity on the measured neutron counts. The temporal changes induced by these factors on the CRNS counting rate were studied and corrected. An example of such studies was by [15], whose work highlights a correction factor to account for the changes in the cosmic ray, and high-energy neutron intensity, which affects the epithermal–fast neutron count measured by a CRNS probe. In addition, [21] developed a correction factor for removing the influence of the temporal changes in atmospheric water vapor content on neutron count. Other studies [15,17,22,24,26] contributed significantly to the correction of uncertainties associated with belowground and aboveground biomass. In the case of temporally stable additional hydrogen pools, estimating their contribution and subtracting it from the neutron counts when converting them into volumetric soil moisture contents reduces uncertainty [15]. According to [17], more complex corrections are required when hydrogen pools change with time. A systematic uncertainty analysis has been conducted by [27], who quantified how the vegetation or soil properties affects the CRNS product. Furthermore, [28] proposed an area-sensitivity function based on the number of neutrons emanating from a given radial distance. All these proposed correction functions formed the basis for standard correcting raw neutron counts to remove uncertainty in soil moisture estimates. These studies focus on reducing uncertainties due to some environmental factors. However, additional hydrogen sources must be accounted for, especially if their contributions change significantly over time to reduce error. It is evident that averaging removed high-peaked noise that we attribute to uncertainty in the count rate. In addition, a few studies investigate the performance of smoothing algorithms on neutron count uncertainty reduction. In some studies, temporal filters such as the moving average and Savitzky–Golay filter were applied to the CRN measurements to reduce the uncertainty in the soil moisture estimates (e.g., Ref. [29]: window length of three measurements; Ref. [30]: window length of seven measurements). Most of these studies apply just one filter with either one or two selected window sizes. Ref. [25], for instance, applied only the moving average filter with window sizes 3 and 9 to reduce the uncertainty before estimating soil moisture. However, studies such as Ref. [31] explored the ability of the moving average and Savitzky–Golay to improve neutron count measurement from the Hydrological Open Air Laboratory (HOAL) in northeast Austria to be used in daily rainfall estimation. In their study, the authors applied the moving average and Savitzky–Golay filter after the standard correction on the neutron count. One question in this context is whether the variables used for standard calibration of neutron count propagate uncertainty to estimated soil moisture. In addition, a limitation of MA and SG filters for filtering neutron count is dependent on the total counts, which are related to the site location (i.e., geomagnetic latitude), elevation, and detector size or type. Another challenge is the ability of the smoothing filters to accurately capture rapid changes in soil moisture due to short-term events such as irrigation or precipitation. For this reason, the dynamic behavior should be better taken into account when reducing the uncertainty of the neutron count using temporal aggregation methods.

In this paper, we explore four smoothing techniques’ ability to optimize the neutron count’s signal-to-noise ratio while maintaining temporal dynamics of soil moisture using synthetic neutron flux created for different geographical locations. The use of synthetic data creates more accurate and scalable surrogate data by adjusting parameters to suit the actual neutron flux measurement. Another advantage of using synthetic data is that the noise added to data is known, while real observations are usually subject to additional noise from various interferences that affect the performance of the correction methods. This method is unreliable for testing filter performance because real signals usually include different noises. Again, synthetic data can be created where actual neutron data are scarce. The study aims to identify an optimal filter and window size, which can return the true value and capture major peaks or patterns in noisy neutron flux measurements. The remainder of the manuscript is organized as follows: Section 2 presents data and the statistical procedure that was used for the analysis. Section 3 presents and discusses the results of four filters at four different sites. Finally, in Section 4, the conclusions and recommendations are given.

## 2. Materials and Methods

### 2.1. Study Site

The study was conducted for four selected Eddy-Covariance (EC) stations in Europe and West Africa, where the prevailing climate presents different dynamics of soil moisture.

Geographically, these stations also demonstrate strong gradients in cut-off rigidities, which is a quantity that describes how Earth’s geomagnetic field shields cosmic-ray particles. The geographical distribution and location of the EC stations are shown in Figure 1. These sites include the West African Science Service Centre on Climate Change and Adapted Land Use (WASCAL) observation network (Gorigo), the GHG-Europe EU-FP7 project (SMEAR II and Conde) and the TERENO (TERrestrial ENvironmental Observatories) test site Rollesbroich. From South to North, the Gorigo site is located in the northern part of Ghana within the Sudan savannah tropical climate zone, and the land surface of this site is a heavily degraded grassland. In Spain, the Conde site is classified as an evergreen broadleaf forest with a Mediterranean climate by the International Geosphere-Biosphere Programme (IGBP). Almost all trees and shrubs remain green year-round. The Rollesbroich test site is a grassland catchment located in the temperate climate zone, western Germany, in the Eifel Mountain range. It covers an area of about 20 ha with altitudes ranging from 474 to 518 masl [32,33]. The Hyytiälä Forestry Field Station of the University of Helsinki, Finland, hosts the SMEAR II (Station for Measuring Forest Ecosystem–Atmosphere Relations) measurement site in boreal homogeneous Scots pine (Pinus sylvestris) stands on flat terrain. Site characteristics are summarized in Table 1.

### 2.2. Data

Hourly soil moisture (SM) for the sites was obtained and used as an input to create the synthetic neutron flux. This ensures that the resulting synthetic neutron flux depicts the temporal dynamics of each site’s SM observation. As a requisite requirement, the neutron flux is always corrected from atmospheric influence such as surface pressure, atmospheric water vapor and incoming neutron. Therefore, surface pressure and absolute humidity data were obtained for each site. These datasets are necessary to standard correct the parameter’s influence on neutron flux when estimating SM. It is also needed to introduce or add the effect of these parameters when creating synthetic neutron flux data from soil moisture. In addition, for the correction of incoming neutron’s influence on the neutron counts, a reference incoming neutron intensity data was obtained from the neutron monitor at Jungfraujoch, which is available via the Neutron Monitor Database (NMDB) at www.nmdb.eu (accessed on 24 March 2022). Data on cut-off rigidity were taken from the Cosmic-ray Soil Moisture Observing System (COSMOS) website (http://cosmos.hwr.arizona.edu/Util/rigidity.php) (accessed on 25 March 2022).

#### Generating Synthetic Neutron Signal for Selected Sites

The major steps for creating the synthetic neutron signal are shown in Figure 2. This study considered only of the soil moisture at 5 cm depth for generating the synthetic neutron data because it is sensitive to irrigation and precipitation events. In addition, the CRNS is highly sensitive to near-surface soil moisture dynamics. The so-called corrected neutron flux *N*${}_{corrected}$, which represents the neutron flux free of atmospheric disturbances, is calculated from the 5 cm soil moisture observations. Based on the [12] proposed relation between gravimetric soil moisture and neutron flux, we estimated the *N*${}_{corrected}$ using Equation (Equation 1):(1)$$N_{corrected}=\left(\frac{0.0808}{\frac{SWC}{\rho_{bd}}+0.115}+0.372\right)\times N_{0},$$
*N*${}_{0}$ represents the calibrated parameter, while $\rho_{bd}$ represents the bulk density at the site. As suggested in different studies [26], *N*${}_{0}$ can be calibrated based on independent soil sampling campaigns. For the purpose of assessing how four different filters reduce uncertainty in CRNS counts, *N*${}_{0}$ was kept constant across all sites. The standard procedure for neutron flux to soil moisture conversion requires the correction of the neutron counts of atmospheric factors that significantly affect the neutron flux. These factors include atmospheric pressure (p), incoming neutron intensity (i) and the absolute humidity (h). The corrected neutron count ($N_{corrected}$, hereafter referred to as $N_{True}$), accounts for the corrections as denoted in Equation (Equation 2):(2)$$N_{corrected}=N_{uncorrected}\times f\left(p,i,h\right),$$
where f(p,i,h) is the atmospheric correction function on neutron counts and $N_{uncorrected}$ is the neutron flux observed by CRNS if no white noise or Poisson noise would occur to the CRNS signal. Therefore, $N_{uncorrected}$ was derived based on Equation (Equation 2). In addition, CRNS are subject to Poisson noise (±n) which is added to the neutron flux $N_{uncorrected}$ to yield the final synthetic neutron flux actually observed by the CRNS at each site:(3)$$N_{syn}=N_{uncorrected}\pm n$$

Figure 2 shows the synthetic neutron (gray) counts generated for the various study sites using its corresponding soil moisture data and atmospheric parameters. The temporal dynamics of the generated synthetic neutron flux capture well the seasonal pattern of soil moisture of the sites.

### 2.3. Analysis

Most studies [29,30] that employ CRNS data resort to moving window filters (e.g., moving average with a window of 24 h). This study used four time-series filters to reduce uncertainty in the generated synthetic neutron signal created for each site. These filters include the moving average (MA), median filter (MF), Savitzky–Golay (SG) and the Kalman filter (KF). The filters present unique ways of removing noise from noisy measurements. An SG filter, for instance, smooths sequential data using least-squares polynomial approximation sliding windows. The polynomial is fitted to a set of input samples and then evaluated at a point within the approximation interval, which is similar to discrete convolution [36,37]. While the median filter estimates values based on the median of the sorted values series of values presented by the window size, Kalman filters recursively estimate the current state using previously estimated states and current measurements. The self-correcting feature of the KF algorithm makes it suitable for improving noisy neutron count measurements. The filters present unique ways of removing noise from noisy measurements. A detailed description of these filters can be found below.

Applying a filter at the right stage of converting neutron count to soil moisture is crucial. Therefore, this study designed two scenarios (A and B) to estimate soil moisture from CRNS. Two scenarios (A and B) were tested for the estimation of soil moisture from the synthetic neutron flux. Thus, for scenario A, the synthetic neutron flux was first corrected for atmospheric influence (pressure, incoming neutron intensity and absolute humidity) before the filtering process. In the case of scenario B, the synthetic neutron data were corrected after filtering. Ultimately, these scenarios also help us determine whether the standard correction process introduces some uncertainty in the soil moisture estimation. A complete summary of the various steps used for the analysis is shown in Figure 3.

#### 2.3.1. Moving Average

The moving average filter remains one of the most common tools for smoothing data, which is often used to capture trends in cyclic statistical surveys. Following the Central Limit Theorem, successive averaging values of a noisy signal should produce a more accurate estimate of the actual signal [38]. Therefore, computer scientists commonly use a moving average (MA) as a filter based on the last *n* data values. A moving average filter is a Finite Impulse Response (FIR) filter used to smoothen the signal from noisy fluctuations. It helps retain the true signal representation or sharp step response. The smoothness of the output signal is determined by the filter’s window length (*w*), which also makes data point transitions sharper. Most simple moving averages are expressed based on Equation (Equation 4) as
(4)$$Y_{MA}\left(n\right)=\frac{1}{w}\sum_{i=0}^{w-1}N\left(t-i\right),$$
where *w* is the filter window length considered for averaging, and N(t) is the synthetic neutron flux of *n* point to be filtered. The window size (*w*) is the only parameter that can be adjusted for this algorithm. A larger window reduces noise while the lag is also increased.

#### 2.3.2. Savitzky–Golay Filter

The Savitzky–Golay (SG) smoothing approach is one of the common methods used for noise filtering [39]. In 1964, Savitzky and Golay proposed the Savitzky–Golay filter as an efficient method for smoothing signals. The Savitzky–Golay method filters noise based on two parameters [39,40]: polynomial order and window size. The core of this algorithm fits a low-degree polynomial in the least squares sense on the samples within a sliding window—the new smoothed value of the center point obtained from convolution. The SG is a particular type of low-pass filter which replaces each value of the time series with a new value obtained from polynomial fit to 2m+1 neighboring points including the point to be smoothed, with *m* being equal to or greater than the order of the polynomial. General expression for the filter can be given as Equation (Equation 5):(5)$$Y_{SG}\left(n\right)=\frac{\sum_{i=-m}^{+m}c_{i}N}{H},$$
where synthetic neutron flux data are indicated with *N*, and the filter coefficient $c_{i}$ is the polynomial of a specific degree that retains higher values. $c_{i}$ and *N* are linearly combined to obtain the smoothed value, *Y*. The convoluting integer *H* equals the smoothing window size (2m+1). In addition, +m and −m denote signal points on the right and left of the current signal point. One of the best advantages of this filter is that it preserves features of the time series such as the maxima and minima. These features are usually flattened by other smoothing techniques, such as moving averages.

#### 2.3.3. Median Filter

The standard median filter is a nonlinear signal processing method capable of removing noise and transients from a signal without distorting the baseline of the time series. In contrast with linear filters, the median filter can remove the effect of extreme noise input values. Median filtering is implemented by allowing a window to slide across the points of a sequence. The data points within the window are sorted in ascending or descending order and replaced with the median of the original values. This produces output data that are often smoother than the original. The input and output (Y${}_{MF}$) of a one-dimensional standard median filter of window size 2m+1 is given as
(6)$$Y_{MF}=med\left\{N(t-1),N(t)\ldots ,N(t+1)\right\}$$
where *Y* is the estimated value at a point in time *t*.

#### 2.3.4. Kalman Filter

Unlike the above-mentioned techniques, Kalman filters make assumptions about the system that generates the signal. It is most commonly used for navigation and tracking when data from different sensors are paired or by using equations. Kalman filters have two models: a process model and a measurement model. The Kalman filter addresses the problem of estimating the state of a discrete-time controlled process governed by the linear stochastic difference equation. The Kalman filter (KF) uses the observed data to learn about the unobservable state variables, which describe the model state. Using the initial state value $N_{syn}^{0}$ and variance $P^{0}$, the prediction equations are expressed as
(7)$$N_{pred}^{t-1}=\Theta N_{syn}^{t-1}$$
(8)$$P^{t-1}=\Theta P^{t-1}\Theta^{\prime }+Q$$

Therefore, the optimal neutron can be estimated using Equation (Equation 9) as:(9)$$Y_{kal}=N_{pred}^{t-1}+K_{t}\left(N_{syn}^{t}-N_{pred}^{t-1}\right)$$
where is the Kalman gain estimated from the predicted variance (*P*). The difference $\left(N_{syn}^{t}-N_{pred}^{t-1}\right)$ in Equation (Equation 9) is the measurement residual, which indicates a discrepancy between the predicted and actual measurements. A residual of zero means that the two are in complete agreement. The Kalman filter improves its prediction based on the Expectation-Maximization (EM) algorithm [41].

#### 2.3.5. Error Measurement

We evaluated the performance of these filters using the Root Mean Square Error (RMSE), Mean Bias Error and Pearson’s correlation coefficient (*r*) [36,42]. Bias and RMSE was used in this study to describe the error’s overestimation/underestimation and magnitude simultaneously. A quantitative measure of the degree of dispersion of predictions could be obtained by both RMSE and MBE [36,43]. The Pearson correlation coefficient is produced via the Pearson product-moment correlation (*r*). This coefficient efficiently measures the degree of linearity between two continuous variables [44]. It can vary from −1 (negative linear relation) to +1 (Positive linear relation). A value zero shows no relationship between the two variables. The error measurements used in this study are the Mean Biased Estimator (MBE), Root Mean Squared Error (RMSE), standard deviation (σ), and Pearson’s correlation (*r*) using Equations (Equation 10)–(Equation 13):(10)$$MBE=\frac{1}{n}\sum_{i=1}^{n}N_{original}-N_{filtered},$$
(11)$$RMSE=\sqrt{\frac{1}{n}\sum_{i=1}^{n}(N_{filtered}-N_{True}}{)}^{2},$$
(12)$$sdev\left(\sigma \right)=\sqrt{\frac{\sum_{i=1}^{n}N_{i}-\bar{N}}{n}},$$
(13)$$r=\frac{\sum \left(N_{filtered,i}-{\bar{N}}_{filtered}\right)\left(N_{True,i}-{\bar{N}}_{True}\right)}{\sum {\left(N_{filtered,i}-{\bar{N}}_{filtered}\right)}^{2}{\left(N_{True,i}-{\bar{N}}_{True}\right)}^{2}},$$
where $\bar{N}$ denotes the mean of the neutron flux (*N*), which is either filtered synthetic neutron ($N_{filtered}$) or true neutron flux $N_{True}$. The *n* represents the neutron flux size considered for the analysis. The efficiency of smoothing filters (MA, MF, SG, and KF) was evaluated using the MBE, RMSE, standard deviation, and Pearson’s correlation coefficient. Different window sizes (6–105 in steps of 6 h) were used to test the performance of three of these filters (MA, MF, and SG). Aside from the filters that operate based on the sliding window technique, the Kalman filter’s output was also evaluated with the error statistics stated above.

## 3. Results

### 3.1. Evaluation of Filters’ Performance at the Four Sites

The application of the Kalman filter (KF) algorithm to the synthetic neutron signal of each site improved the signal-to-noise ratio. Figure 4 shows the statistical measure of the performance of Kalman-filtered synthetic neutron signal for the four study sites. According to the results, a robust linear agreement (correlation) was observed at SMEAR II (0.91), Gorigo (0.95), Rollesbroich (0.98) and Conde (0.97), as indicated in Figure 4d. To this effect, the KF reduced the standard deviation of the synthetic neutron flux at SMEAR II (26.34 cph), Gorigo (52.07 cph), Rollesbroich (37.99 cph), and Conde (39.74 cph) by 52.05 %, 22.61 %, 28.38 % and 26.52 %, respectively. Figure 4b presents the σ obtained after applying KF. The standard deviations for the true neutron flux, the synthetic signal, and the KF-filtered neutron signal are given in Table 2. These results show a significant reduction of the Poisson noise introduced during the signal creation. A possible explanation for this might be that the KF filter implements the Expectation Maximization algorithm and consists of two iterative steps (Kalman smoother and maximization of the expected log-likelihood). This reduces the variance in its predictions. More than a single run (in this study, 100 iterations of the EM algorithm) is required to improve the noise estimation sampled from a given distribution; therefore, every simulation run results in different state estimates [45].

When the number of iterations increases, the Kalman gain decreases, decreasing the signal error [41]. Although KF showed good uncertainty reduction results, a negative bias was observed at Gorigo (1.72 cph) and Rollesbroich (2.63 cph). Whereas, at Conde (0.31 cph) and SMEAR II (2.64 cph) sites, a positive bias was observed (see Figure 4a). KF underestimated at Conde and SMEAR II, while it overestimated at Rollesbroich and Gorigo.

Moreover, the remaining filters (MA, MF and SG), which use the principle window size as part of their algorithm, also showed significant results. Figure 5 shows the MA, MF and SG performance for different window sizes at the four sites. The correlation pattern between the filtered synthetic signal by various filters and the true neutron count increased with the window size increasing to an optimal window size. At the same time, RMSE and σ for all filters decreased as the window size increased to an optimal window size. Observations of SG and MA concerning the window size corroborate with [31], where similar results were reported when they evaluated these filters for different window sizes (less than 24 h). In addition, similar behavior of the filter performance as a function of the window size was demonstrated by [46]. The correlation values observed for the SMEAR II site for all filters at different window sizes ranged between 0.90 and 0.98, as shown in Figure 5d. This correlation range observed varies at all sites: at Gorigo (0.80–0.96), Rollesbroich (0.75–0.98) and Conde (0.90–0.99) sites for the different window sizes tested in this study (see Figure 5h,l,p).

These are expected results because these algorithms are implemented on data points to estimate smoothed values; the number of data points that was considered in fitting increases as window size increases. This yields a better polynomial result especially with SG filters, which corresponds with RMSE and MBE decreasing, and better performance is acquired. In addition, for MF and MA, when the filter window size is too small, details of neutron count events occurring at specific periods are lost. On the other hand, irrelevant information (outliers) is maintained for the new neutron count value estimation if it is too large. These filters will result in different soil moisture/neutron fluxes, which is mainly due to the window size (MA and MF) and polynomial order (SG filter). The maximum and minimum window points of the correlation and RMSE, respectively (see RMSE and correlation plot in Figure 5), could be regarded as the trade-off or optimal point. Therefore, this implies that the trade-off point will better estimate the neutron counts closer to the actual value. In terms of the filters’ performance to window sizes, the correlation pattern observed for all sites at smaller window sizes was such that MA > MF > SG-3 > SG-4. However, when the window size increased, the performance was SG-4 > SG-3 > MF > MA. It is important to note that a small window size will perform better with low noise density than with high noise density [47]. Large window size limits the ability of these filters to follow rapid but relevant changes in the synthetic neutron counts. On the other hand, a narrow window may cause the results to over-fit the time series, since it retains some of the noise.

Thus, for accurate determination of relevant peak signals, the critical factor is the number of data points smoothing the window size, which are located in the half-width of the peak. As the number of data points at both sides of the peak is large, the value of the peak point is reduced. Although this improves filtering, it also causes a reduction in the amplitude, especially if the peaks are relevant. Another parameter that improves error reduction is the polynomial order of the SG filter. We can, therefore, always approximate data inside a window with a low-order polynomial once we set its width. For sharper peaks, this might be different. Increasing the polynomial’s order would be the best solution. If the order is kept the same (without expanding the window width), the polynomial would closely mimic the noise oscillations. However, the signal is unsmoothed if the polynomial order equals the window length (n=w): the polynomial order (n) uses *w* window length to estimate the polynomial coefficients. Therefore, if n=w, smoothing is not performed, but the polynomial interpolates the data points [48,49]. Consequently, choosing a polynomial order less than the window length is always necessary. Our results also suggest that as the polynomial order of the SG filter increases, the optimal window size becomes relatively large, as observed between SG-3 and SG-4 correlation for all sites. Thus, the trade-off or optimal point between the window size and polynomial order where errors are minimal occurs at a shorter window for the 3rd-order polynomial compared to the 4th-order polynomial SG variant. Despite strong positive correlations for all filters at all window sizes, the MA, MF and SG filters underestimated the original neutron flux at SMEAR II (MBE; −0.55 to −0.39 cph), Rollesbroich (MBE; −0.50 to −0.19 cph) and Conde (MBE; −3.0 to −0.90 cph). At Gorigo, the MF filter overestimated the original neutron flux for all window sizes tested (MBE; 0.50 to 1.20 cph). Meanwhile, the SG and MA filters showed underestimation with mean bias error ranging between −0.50 and 0.01 cph for all window sizes. The results for the SG filter indicate a slight difference in output as the polynomial order increased, which is similar to the results reported by [50].

### 3.2. Optimal Filter and Window Length

A filter’s robustness and efficiency lies in its ability to preserve relevant peaks or slopes from the synthetic neutron flux that appear in the actual neutron flux. Figure 6 shows the time series of the reconstructed synthetic neutron flux from the SG filters, MA, MF, and KF. Each filter depicted a similar pattern as the sites’ true neutron flux ($N_{cor}$). Additionally, the most considered smoothing technique (MA 24-h window size) in CRNS was included in the analysis of this section. Results showed that the optimal window size for MA, MF, SG-3 and SG-4 varied from site to site (see Table 3).

Selection of the best window size for MA, MF, and SG depended on the correlation, mean bias error and the standard deviation assessed in Section 3.1. At various sites, KF, MA, MF, SG and MA-24 filters captured the seasonal pattern of the true neutron flux at all sites, as shown in Figure 6. On the other hand, the 24 h MA results showed some delay in periods of sharp changes and a slow change rate in neutron flux (Figure 6b,c). The window size tuning parameter exhibits effects such as reducing noise and lag introduction. The small window size retains some amount of noise while following rapid changes. As the window size increases, noise reduction improves, but lags are introduced into the result (especially the MA filter). Therefore, filtered synthetic neutron flux has a lower bias, RMSE, and standard deviation than the original neutron flux. The optimal window size obtained in this study reflects the trade-off between large and small window sizes. These results are more helpful in filtering data for short-term or long-term soil moisture changes to suit hydrological purposes, e.g., resulting from irrigation, drought, precipitation and snowmelt.

Since these are important hydrological processes, a filtered CRNS should be able to capture such sharp soil moisture changes as well.

Therefore, two selected cases of sharp neutron (converted to soil moisture) changes at Gorigo and Rollesbroich were used to test the performance of the used filters (Figure 7a–d). To this end, a comparison between the actual change observed from the true soil moisture and the soil moisture estimated using all filters (MA, MF, SG, and KF) was performed using the optimal windows identified at these two sites. In addition to using the optimal windows identified at these two sites, we tested the performance of the filters (MA, MF, SG3 and SG4) at a window size of 12 h. During the event period (pale red shaded region in Figure 7), smaller windows (Gorigo: SG3-12 h, Rollesbroich: SG3-12 h) captured the sharp changes in soil moisture more closely. Generally, as expected, the smaller window size (12 h) captured the short changes in the soil moisture better, in which the SG3-12 h filter produced the lowest relative percentage difference at Gorigo (17.08%) and at Rollesbroich (12.04 %) as indicated in Figure 7b,d respectively. These results confirm that the 12-h filters can follow rapid changes quite well, but, on the other hand, they performed worst in case of moderate soil moisture variations during a more extended period and thus failed to capture the long-term trend. This result implies that the optimal filter type and window size may also depend on soil moisture application, i.e., 12-h filters should be used in case of studies in which rapid soil moisture changes are expected, e.g., flood forecast or irrigation management. In contrast, filters with larger window sizes should be employed in case of long-term water balance studies.

A further assessment of the performances of the optimized filter results for the MA, MA-24 h, MF, SG and KF for the individual sites is presented in Figure 8. All filters underestimated the true neutron flux at SMEAR II, Gorigo and Rollesbroich sites (Figure 8). Nevertheless, the Kalman filter performed best at SMEAR II with a standard deviation (12.62 cph) close to $N_{True}$ (13.29 cph) among the filters followed by the SG-4, SG2/SG3 and MA, as presented in Figure 8a.

For the Gorigo data, the MF showed the best performance. The filter showed the highest correlation (0.97), least RMSE (11.25 cph) and a standard deviation (41.34 cph) similar to the true neutron counts (see Figure 8b). Again, the Kalman filter presented the best results at both Rollesbroich and Conde. Overall, the Kalman filter showed great improvement in reducing uncertainties in the neutron flux to preserve important features similar to the true neutron flux. As expected, all filters whose algorithm uses filtering windows show a reduction in the measurement uncertainty with increasing window size to some point. For most filters, the amount of noise reduction is proportionally accompanied by signal degradation [40], which may distort the filtered signal.

### 3.3. Uncertainty Propagation from CRNS Standard Correction

Soil moisture estimates from cosmic ray neutrons are also liable to the systematic error of atmospheric parameters (absolute humidity, pressure, temperature and incoming neutron intensity) used in the correction of the neutron counts. The procedure of neutron count data pre-processing prior to noise reduction also helped improve the error reduction in cosmic neutron counts. Figure 9 shows the effect of applying the various filtering techniques after (Scenario A) and before (Scenario B) the standard atmospheric correction on neutron counts. Soil moisture is improved when the synthetic neutron flux was corrected before filters were applied, as shown in Figure 9a. In contrast, estimated soil moisture shows a random fluctuation compared to the observed soil moisture (Figure 9b) when the standard correction was performed after the filtering process. Parameters used for standard correction (absolute humidity, incoming neutron intensity and surface pressure) could be the source of these random fluctuations. This improvement in the soil moisture estimates is expressed with the RMSE, as indicated in Table 4.

The results indicate that the correction parameters are also prone to some amount of uncertainty that is propagated to soil moisture during the correction of neutron flux. These uncertainties may influence hydrological modelling and soil moisture representation in forecasting models.

## 4. Conclusions

Accurate soil moisture estimates are essential for crop management planning, calibration of hydrological models and improving extreme event forecasts. Therefore, it is imperative to enhance existing soil moisture estimation techniques by reducing uncertainties in their measurement.

This study aimed to assess the optimal approach for reducing cosmic ray neutron count uncertainties using four different filtering algorithms. The investigation of filter performance at different sites has shown that the Kalman filter is robust in uncertainty reduction. The KF presented a unique algorithm which maintained relevant sharp changes in the neutron count at three locations. In addition, the evaluation of Moving Average (MA), Median filter (MF) and Savitzky–Golay (SG) filters revealed that the choice of optimal window size changes based on the level of noise present in the synthetic data. One interesting result emerging from this study is that a relatively large window size captures the long-term pattern in soil moisture well. In contrast, shorter window sizes capture short-term events well, especially in combination with the Savitzky–Golay filter. Therefore, short window sizes should be used for analyzing short-term events such as irrigation, infiltration, or snowmelt. In contrast, a larger window size is more suitable for studying long-term hydrological events, such as droughts.

The second significant finding was that the variable used for the standard correction also introduced errors in CRNS soil moisture estimates. Therefore, it is recommended that the CRNS count be filtered after standard correction before further analysis. However, one limitation of the current study was to assess filter performance on a seasonal basis. We, therefore, caution that, when applying these filters, identifying window size and interpreting sub-daily soil moisture data from CRNS, care should be taken. We anticipate that our study’s results will help create better awareness of CRNS data filtering and reduce uncertainty in CRNS soil moisture product.

## Figures and Tables

**Figure 1 sensors-22-09143-f001:**
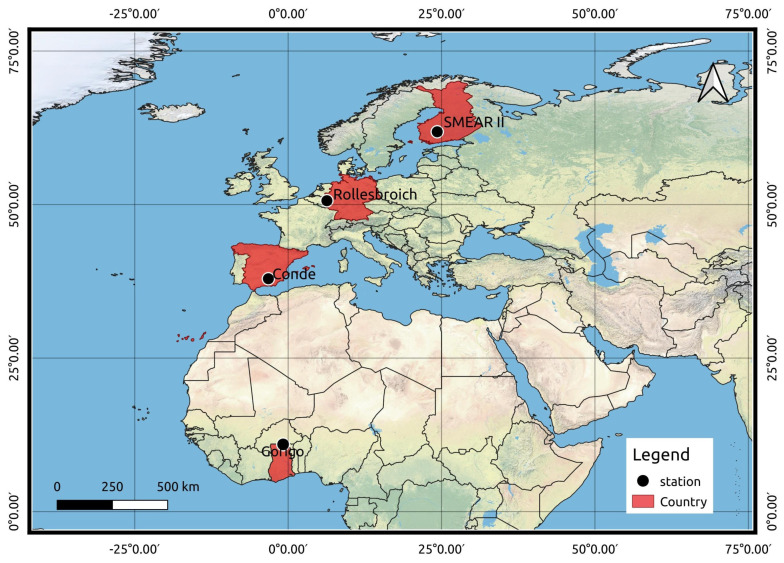
Map showing geographical location of stations used in the study. The countries where stations are located are highlighted in red.

**Figure 2 sensors-22-09143-f002:**
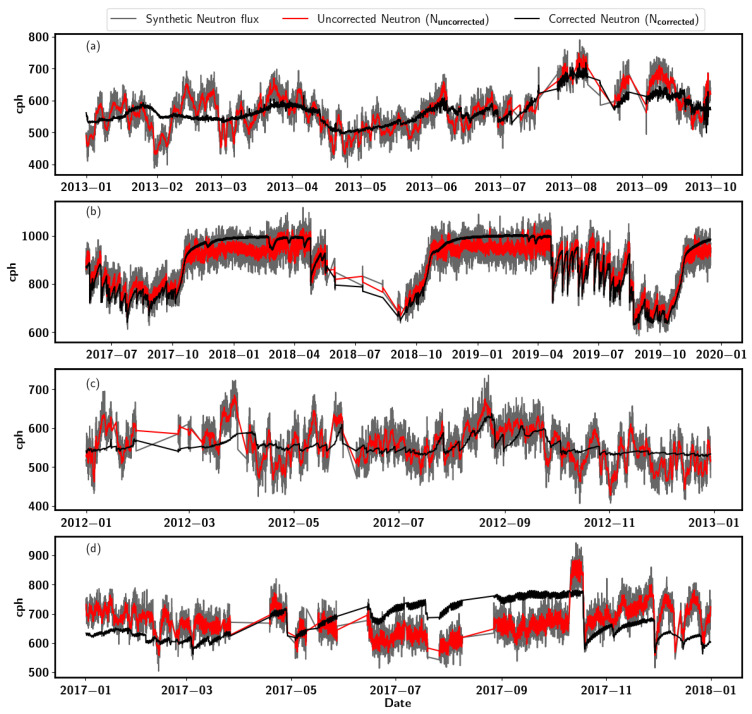
Graphical representation of synthetic neutron flux $N_{syn}$ (gray) generated for Finland (**a**), Gorigo (**b**), Rollesbrioch (**c**) and Conde (**d**). The black line represents the neutron flux $N_{True}$ only subject to the soil moisture change, while red is the uncorrected neutron flux $N_{uncorrected}$ subject to atmospheric factors and soil moisture.

**Figure 3 sensors-22-09143-f003:**
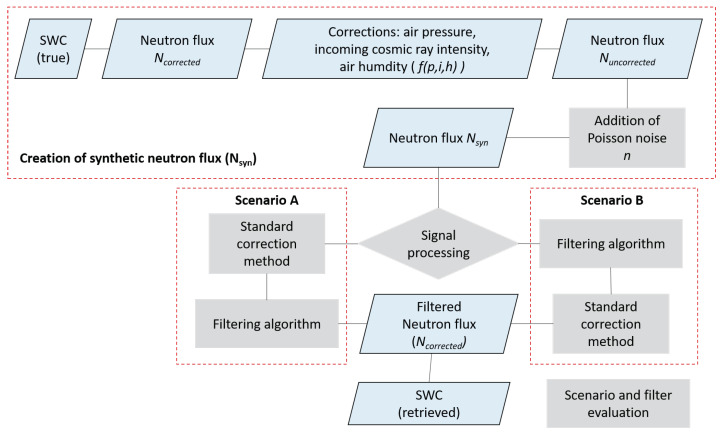
Schematic flow for generating synthetic neutron flux. Key steps for analysis of two scenarios when filters are applied after (scenario A) or before (scenario B) standard correction of synthetic neutron signal.

**Figure 4 sensors-22-09143-f004:**
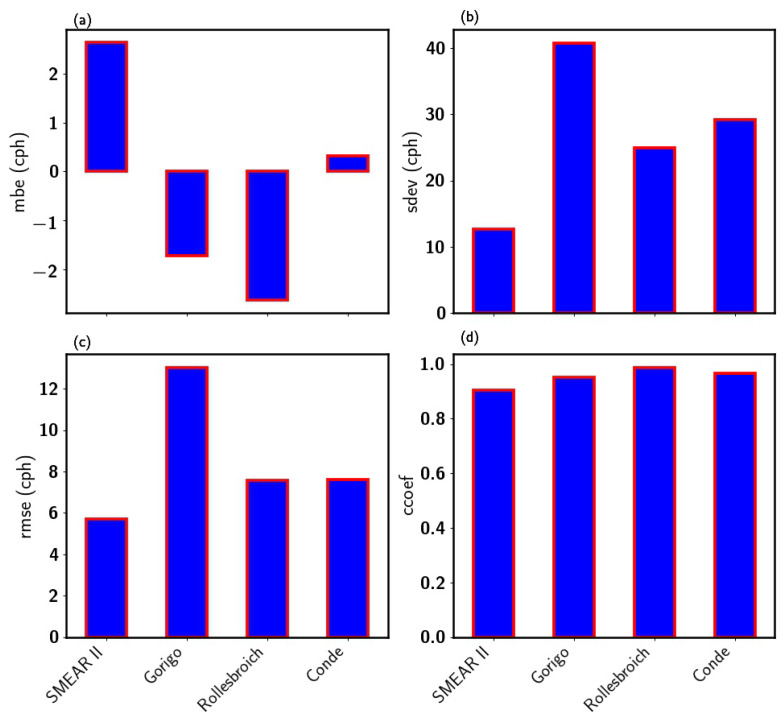
Performance of Kalman filter at the different station based on the MBE (**a**), standard deviation (**b**), RMSE (**c**) and correlation (**d**).

**Figure 5 sensors-22-09143-f005:**
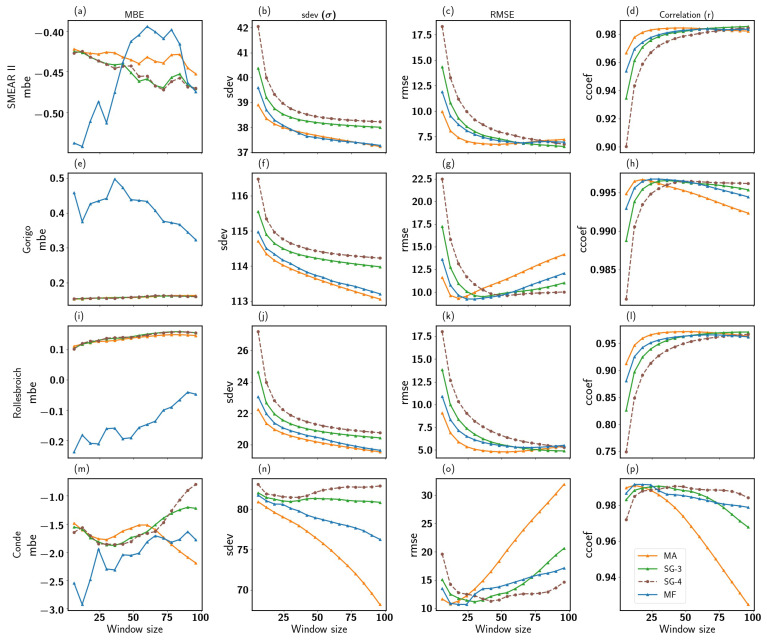
Performance of MA (red), MF (blue) and SG filters as a function of window length. The MBE, standard deviation, RMSE and correlation are presented in columns for SMEAR II (**a**–**d**), Gorigo (**e**–**h**), Rollesbroich (**i**–**l**) and Conde (**m**–**p**) for the four stations (in rows), respectively.

**Figure 6 sensors-22-09143-f006:**
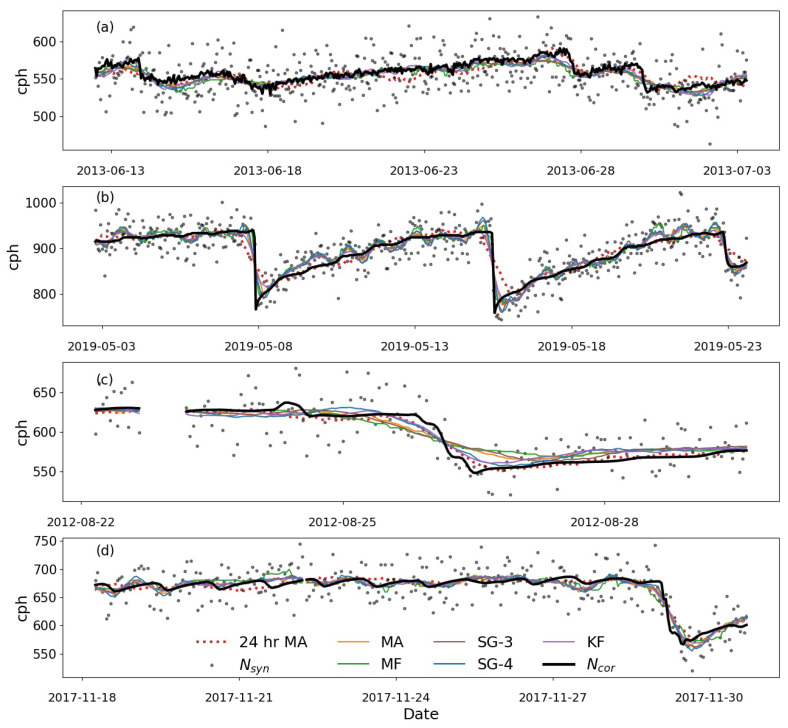
Time series of synthetic neutron flux (gray dot) and filtered flux using the MF, MA, SG and KF filtering techniques at SMEAR II (**a**), Gorigo (**b**), Rollesbroich (**c**) and Conde (**d**) sites. The black line represented the original neutron flux.

**Figure 7 sensors-22-09143-f007:**
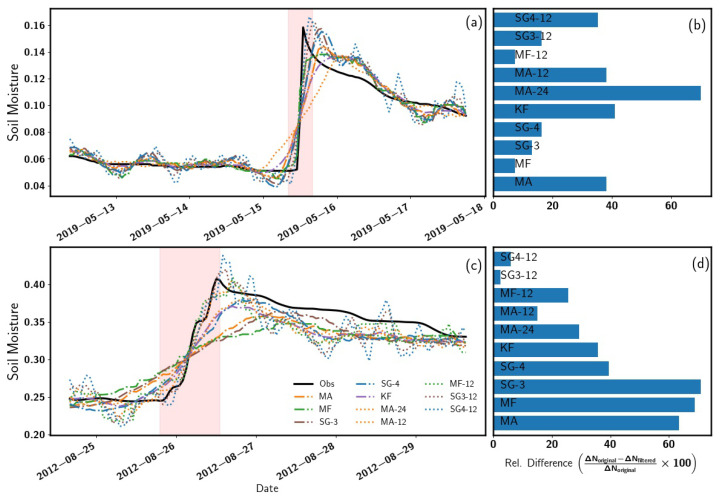
Relative percentage difference (**b**) and (**d**) of filter response to shape changes in soil moisture at Gorigo (**a**), Rollesbroich (**c**) sites, respectively. The pink shaded region represents the interested event.

**Figure 8 sensors-22-09143-f008:**
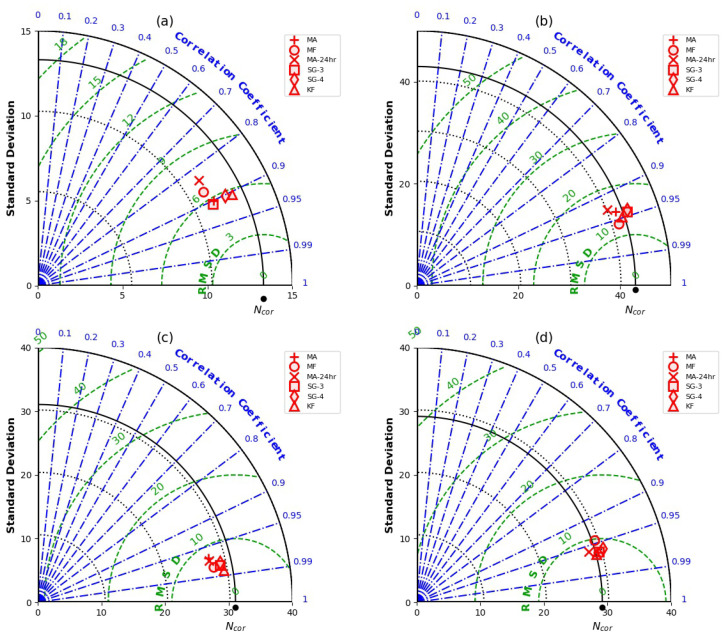
Performance of the optimal window size of the different filtering approaches at SMEAR II (**a**), Gorigo (**b**), Rollesbroich (**c**) and Conde (**d**). The original neutron flux is denoted by the black dot.

**Figure 9 sensors-22-09143-f009:**
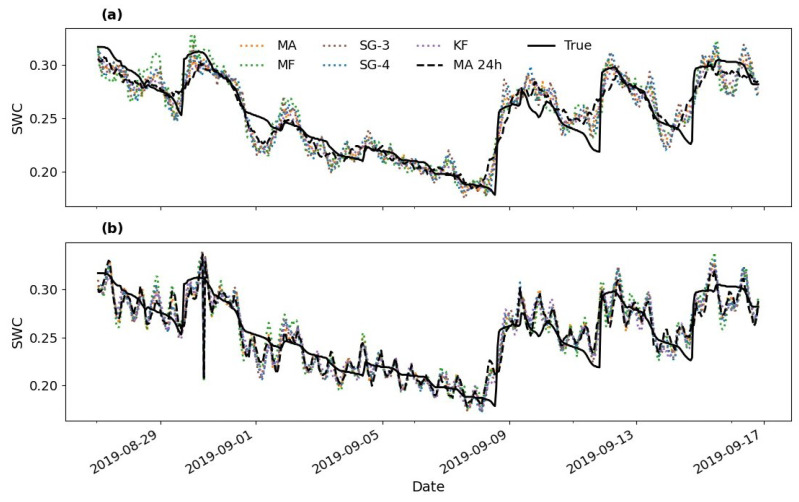
Time series of observed soil moisture (black line) at Gorigo and estimated soil water content from filtered synthetic neutron count based on two scenarios. Scenario A (**a**): Filters applied after atmospheric correction. Scenario B (**b**): Filters applied before atmospheric correction.

**Table 1 sensors-22-09143-t001:** Summary of the site characteristics.

Station	Lon/Lat	Bulk Density (g/cm${}^{3}$)	Rigidity Cut-Off (GV)	Other Site Information
Gorigo	0.82/10.93	1.54	14.68	Highly degraded grassland
				Loamy sand soil
Rollesbroich	6.30/50.63	1.09	3.27	Managed grassland
				Silty clay loam
SMEAR II	24.29/61.84	0.85	1.11	Homogenous Scots pine trees
				Silty sand [34]
Conde	−3.22/37.91	1.37	8.33	Evergreen trees and shrubs.
				Clayey loam [35]

**Table 2 sensors-22-09143-t002:** Standard deviations of true neutron flux, the synthetic signal and the Kalman-filtered signal for the four sites.

Station	True Neutron	Synthetic Neutron	KF-Filtered Neutron
Gorigo	43.05	52.07	40.30
Rollesbroich	31.16	37.99	29.99
SMEAR II	13.31	26.34	12.63
Conde	29.24	39.74	29.20

**Table 3 sensors-22-09143-t003:** Optimal window size (h) of various filters for the different sites.

Station	MA (h)	MF (h)	SG-3 (h)	SG-4 (h)
Gorigo	30	36	78	84
Rollesbroich	18	18	30	48
SMEAR II	36	42	54	84
Conde	18	12	30	36

**Table 4 sensors-22-09143-t004:** RMSE comparison of scenarios A and B regarding soil moisture uncertainty reduction.

Filter	Scenario A (cm${}^{3}$/cm${}^{3}$)	Scenario B (cm${}^{3}$/cm${}^{3}$)
KF	0.006	0.008
MA (30 h)	0.006	0.009
SG-3 (78 h)	0.007	0.009
SG-4 (84 h)	0.007	0.008
MA (24 h)	0.007	0.009
MF (36 h)	0.007	0.009

## Data Availability

The data supporting the study finding are available online. Finland (FI-Hyy) and Spain (ES-Cnd) can be access at (http://www.europe-fluxdata.eu/home/ (accessed on 4 March 2022)). Contact Samuel Sapanbil Guug (guug.s@wascal.org) for Gorigo data.

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
