# Peer review of "Optimal Temporal Filtering of the Cosmic-Ray Neutron Signal to Reduce Soil Moisture Uncertainty"

_sensors, 2022, doi:10.3390/s22239143_

Round 1

Reviewer 1 Report

Please find the attachment for comments to authors.

Author Response

Response to Reviewer 1 Comments

Dear Editor,

Thank you for the opportunity given us to publish our work in your journal.

We also thank the reviewers for their valuable comments and suggestions meant to improve our manuscript. The reviewer’s comments are in black and our responses to the comments are in red.

Comment 3: Overall, the paper suffers from too many grammatical issues at this time. These issues prevent a productive review process from occurring at this time. The authors are encouraged to overhaul their writing and resubmit a more cogent manuscript with a better-articulated narrative.

Response 3: We sincerely appreciate the reviewer’s comments. We have proofread the manuscript, corrected the grammatical errors, and misplaced punctuations, and reconstructed sentences where necessary for clarity. We are very confident that the overall grammar in the paper has significantly improved.

Comment 4: Why uncertainty analysis is not being used in the present study? Is there no sort of problem with the output from the datasets? Uncertainty analysis/bias correction is necessary if the output being used is incompatible with the analysis. I do not believe this is the case here. Also, many uncertainty analysis methods destroy the physical relationship between variables and climate change physics, and this begs the question, why use climate variables datasets at all?

Response 4: Thank you very much for your comments.

The present study did not include uncertainty analysis/bias correction, it focuses on reducing the uncertainty associated with the CRNS counting rate. The uncertainty of CRNS neutron count rates follows Poisson statistics (Knoll, 2000; Zreda et al., 2012), where the uncertainty is equal to the square root of the total counts measured.

This study aims at assessing filters’ performance in reducing counting rate uncertainty. Therefore a synthetic neutron flux was created with known uncertainty (Poisson noise).

The climate dataset (absolute humidity, surface pressure, soil moisture) was used to create synthetic neutron flux data. The synthetic data mimics the seasonal pattern of the site’s neutron count assuming it was measured with the cosmic ray neutron probe. In order to assess the performance of the filtering methods employed in our study, each filter’s ability to remove the known uncertainty added to the synthetic data was assessed. Besides, these improved filtered neutron counts improve soil moisture estimated from the filtered CRNS neutron count data.

Comment 5: Introduction and discussion, please extend your argument and discuss the recent global warming in terms of soil moisture, air temperature, and relative humidity, and their physical mechanism contribution towards Cosmic-Ray Neutron Signal.

Response 5: Thank you very much for your insightful comment.

We have added a brief explanation of global warming's impact on soil moisture which affects food security. In addition, more information regarding soil moisture, air temperature, and relative humidity effect on Cosmic-Ray Neutron Signal has been provided in lines 54-73 and is not necessarily global warming. It will be interesting to note that, the effect of climate change on these parameters will still have the same influence on the cosmic ray neutron counts as described in some studies (Zreda et. al, 2012; Bogena et. al, 2013; Rosolem et al. 2013a; Baatz et. al, 2015).

However, It will be very interesting to investigate global warming on neutron measurement, which is not the focus of the current study. The current study rather focuses on reducing the uncertainty associated with Cosmic-Ray Neutron Sensors' counting statistics.

Reviewer 2 Report

1.     The manuscript is concerned with optimal temporal filtering of the cosmic-ray neutron signal to reduce soil moisture uncertainty. It is relevant and within the scope of the journal.

2.     However, the manuscript, in its present form, contains several weaknesses. Adequate revisions to the following points should be undertaken in order to justify recommendation for publication.

3.     Full names should be shown for all abbreviations in their first occurrence in texts. For example, SMEAR II in p.1, COSMOS in p.4, etc.

4.     For readers to quickly catch the contribution in this work, it would be better to highlight major difficulties and challenges, and your original achievements to overcome them, in a clearer way in abstract and introduction.

5.     It is shown in the reference list that the authors have several publications in this field. This raises some concerns regarding the potential overlap with their previous works. The authors should explicitly state the novel contribution of this work, the similarities, and the differences of this work with their previous publications.

6.     p.1 - cosmic ray neutron sensor is adopted to determine field-scale soil moisture. What are the other feasible alternatives? What are the advantages of adopting this approach over others in this case? How will this affect the results? The authors should provide more details on this.

7.     p.1 - moving average, median filter, Savitzky-Golay filter, and Kalman filter are adopted to reduce neutron count error while preserving the temporal SM dynamics as best as possible. What are the other feasible alternatives? What are the advantages of adopting these techniques over others in this case? How will this affect the results? More details should be furnished.

8.     p.3 - four selected Eddy-Covariance stations in Europe and West Africa are adopted as case studies. What are other feasible alternatives? What are the advantages of adopting these case studies over others in this case? How will this affect the results? The authors should provide more details on this.

9.     p.4 - hourly soil moisture, air temperature, relative humidity, and atmospheric pressure data are adopted in the analysis. What are the other feasible alternatives? What are the advantages of adopting these parameters over others in this case? How will this affect the results? More details should be furnished.

10.  p.4 - 1020 is adopted for the calibrated parameter for all sites. What are the other feasible alternatives? What are the advantages of adopting this value over others in this case? How will this affect the results? The authors should provide more details on this.

11.  p.5 - two scenarios are adopted for the introduction of uncertainty. What are the other feasible alternatives? What are the advantages of adopting these scenarios over others in this case? How will this affect the results? More details should be furnished.

12.  p.6 - the schematic flow as shown in Figure 3 is adopted for generating synthetic neutron flux and major steps for analysis. What are other feasible alternatives? What are the advantages of adopting this schematic flow over others in this case? How will this affect the results? The authors should provide more details on this.

13.  p.7 - four error measurements are adopted to evaluate the performance of thefilters. What are the other feasible alternatives? What are the advantages of adopting these evaluation metrics over others in this case? How will this affect the results? More details should be furnished.

14.  p.7 - specific window sizes are adopted to test the performance of three of those filters (MA, MF, and SG). What are the other feasible alternatives? What are the advantages of adopting these window sizes over others in this case? How will this affect the results? More details should be furnished.

15.  p.13 - “…However, relatively large window size based on the degree of variance in data may result in the overestimation of the correct neutron flux measurement.…” Some justification should be furnished on this issue.

16.  Some key parameters are not mentioned. The rationale on the choice of the set of parameters should be explained with more details. Have the authors experimented with other sets of values? What are the sensitivities of these parameters on the results?

17.  The discussion section in the present form is relatively weak and should be strengthened with more details and justifications.

18.  Some assumptions are stated in various sections. More justifications should be provided on these assumptions. Evaluation on how they will affect the results should be made.

19.  Moreover, the manuscript could be substantially improved by relying and citing more on recent literature about real-life applications of modeling techniques in soil moisture estimation such as the following. Discussions about result comparison and/or incorporation of those concepts in your works are encouraged:

          Breen, K.H., et al., “A Hybrid Artificial Neural Network to Estimate Soil Moisture Using SWAT+ and SMAP Data,” Machine Learning and Knowledge Extraction 2(3): 283-306 2020.

          Liu, Y.X.Y., et al., “Generating high-resolution daily soil moisture by using spatial downscaling techniques: a comparison of six machine learning algorithms,” Advances in Water Resources 141: 103601 2020.

          Moazenzadeh, R., et al., “Soil Moisture Estimation Using Novel Bio-inspired Soft Computing Approaches,” Engineering Applications of Computational Fluid Mechanics 16 (1): 826-840 2022.

20.  Some inconsistencies and minor errors that needed attention are:

          Replace “…Different window size (6 – 105 in steps of 6 hrs) were used…” with “…Different window sizes (6 – 105 in steps of 6 hrs) were used …” in lines 189-190 of p.7

          Replace “…We therefore, caution that when applying these filters, identifying window size and interpreting sub-daily soil moisture data from CRNS care should be taken…” with “…We therefore caution that, when applying these filters, identifying window size and interpreting sub-daily soil moisture data from CRNS, care should be taken …” in line 332 of p.14

21.  In the conclusion section, the limitations of this study, suggested improvements of this work and future directions should be highlighted.

Author Response

Response to Reviewer 2 Comments

Dear Editor,

Thank you for the opportunity given us to publish our work in your journal.

We also thank the reviewers for their valuable comments and suggestions meant to improve our manuscript. The reviewer’s comments are in black and our responses to the comments are in red.

Comment 3: Full names should be shown for all abbreviations in their first occurrence in texts. For example, SMEAR II in p.1, COSMOS in p.4, etc.

Response 3: Thank you very much for highlighting the omission of the abbreviation's full names. The current manuscript addresses the full names of the abbreviations. We have changed in the current manuscript on lines 9 and 140 for the abbreviations SMEAR II and COSMOS respectively.

Comment 4: For readers to quickly catch the contribution in this work, it would be better to highlight major difficulties and challenges, and your original achievements to overcome them, in a clearer way in the abstract and introduction.

Response 4: Thank you for the suggestion. Lines 1-9 point out the problem this study seeks to answer and the methods selected to address the identified challenge. However, lines 95-113 have been improved in the current manuscript, highlighting some challenges and limitations of other studies and how this present study addresses them.

Comment 5: It is shown in the reference list that the authors have several publications in this field. This raises some concerns regarding the potential overlap with their previous works. The authors should explicitly state the novel contribution of this work, the similarities, and the differences of this work with their previous publications.

Response 5: Thank you very much. As suggested, in the current manuscript, lines 84-101 we discussed previous works which are similar to our study and highlighted some differences. One of such could be seen on page 3 lines 93 – 95 reads, ‘One question in this context is whether the variables used for standard calibration of neutron count propagate uncertainty to estimated soil moisture. This highlights a gap in the previous study which we will address in this current study. Also, lines 101-113 provide a detailed description of what this recent work proposes.

Comment 6: p.1 - cosmic ray neutron sensor is adopted to determine field-scale soil moisture. What are the other feasible alternatives? What are the advantages of adopting this approach over others in this case? How will this affect the results? The authors should provide more details on this.

Response 6: Thank you very much for your suggestions.

In the previously submitted manuscript, lines 28-29 discussed other feasible alternatives for soil moisture estimation. However, in lines 31-56 of the current manuscript, we have made considerable effort to enhance the discussion on soil moisture measurement techniques with their capabilities.

Comment 7: p.1 - moving average, median filter, Savitzky-Golay filter, and Kalman filter are adopted to reduce neutron count error while preserving the temporal SM dynamics as best as possible. What are the other feasible alternatives? What are the advantages of adopting these techniques over others in this case? How will this affect the results? More details should be furnished.

Response 7: Thank you. There are other filters, but the moving average filter is the most common in several CRNS filtering studies. We discussed these in the introduction, lines 83- 92. Moving Average was included to compare results to the Median filter (MF), Savitzky-Golay (SG) and the Kalman filter (KF) other three filters (MF, SG and KF). The median filter can deal with edge problems. SG captures the maximum and minimum points well. In contrast, the KF filter uses iterative steps to improve prediction. All has been clearly captured in the current revision on lines 177 – 190.

Point 8: p.3 - four selected Eddy-Covariance stations in Europe and West Africa are adopted as case studies. What are other feasible alternatives? What are the advantages of adopting these case studies over others in this case? How will this affect the results? The authors should provide more details on this.

Response 8:

Thank you very much for the suggestion.

The choice of the selected EC sites in the current study was limited by data availability. However, these sites are located in different climate zone. This also presents the opportunity to experiment with sites with different cut-off rigidity. The cut-off rigidity describes the number of cosmic ray particles (neutron fluxes) received at the earth's surface. As indicated by Kress et al., 2015, the cut-off rigidity reduces with increasing latitude, influencing the amount of neutron flux at the surface. Again these sites are characterized by different climate and surface properties which affect the soil moisture dynamics.

Point 9: p.4 - hourly soil moisture, air temperature, relative humidity, and atmospheric pressure data are adopted in the analysis. What are the other feasible alternatives? What are the advantages of adopting these parameters over others in this case? How will this affect the results? More details should be furnished.

Response 9:

The adopted parameters are critical in converting cosmic ray neutron counts to soil moisture. The relation between soil moisture and neutron flux by Desilets et al. (2010) requires these parameters. Therefore, we created the synthetic neutron flux based on this formulation with our soil moisture and correction parameters (pressure, air temperature, relative humidity and incoming neutron intensity). Note that the temperature and relative humidity were used to estimate absolute humidity, where data is unavailable.

In addition, meteorological parameters help create synthetic neutron flux data that mimics the actual temporal pattern of soil moisture and other variables that influence neutron flux.

This was admittedly unclear in the previously submitted manuscript. We have revised the Data section on lines 125 to 136.

The present manuscript reads, "Hourly soil moisture (SM) for the sites was obtained and used as an input to create the synthetic neutron flux. This ensures that the resulting synthetic neutron flux depicts the temporal dynamics of each site's soil SM observation. As a requisite requirement, the neutron flux is always corrected from atmospheric influence such as surface pressure, atmospheric water vapour and incoming neutron. Therefore, surface pressure, absolute humidity and incoming neutron flux were obtained for each site. These datasets are necessary to standard correct the parameter's influence on neutron flux when estimating SM. It is also needed to introduce or add the effect of these parameters when creating synthetic data from soil moisture."

Furthermore, a detailed explanation of how the synthetic flux can estimate soil moisture is on page 3, lines 137-172 (Figure 3). Hence, the reasons for the choice of parameters.

Point 10: p.4 - 1020 is adopted for the calibrated parameter for all sites. What are the other feasible alternatives? What are the advantages of adopting this value over others in this case? How will this affect the results? The authors should provide more details on this.

Response 10: Another feasible alternative is obtaining calibration parameters for each site, which is only possible for some stations. However, a series of tests conducted on different calibration parameters showed no significant impact. Also, the calibration parameter does not affect the counting statistics' uncertainty.

Point 11: p.5 - two scenarios are adopted for the introduction of uncertainty. What are the other feasible alternatives? What are the advantages of adopting these scenarios over others in this case? How will this affect the results? More details should be furnished.

Response 11:

Thank you for drawing our attention to this statement which misinforms what is described and presented by the flow chart (Fig 3) depicts. This mistake has been rectified in the current manuscript on lines 187-188, which reads.

- this study designed two scenarios (A and B) to estimate soil moisture from CRNS counts.

In these two scenarios, the difference is the stage or step where the synthetic neutron signal is filtered. Thus for Scenario A, the processed signal (the generated synthetic neutron flux) is corrected from the atmospheric and surface factors influencing neutron flux before applying filters. The synthetic neutrons are filtered in scenario B before the standard neutron count correction is applied. These two scenarios are tailored for this current study. It provides the opportunity to understand whether uncertainties are introduced to CRNS-estimated soil moisture when the standard correction of cosmic ray neutron flux is performed before/after filtering.

Point 12: p.6 - the schematic flow as shown in Figure 3 is adopted for generating synthetic neutron flux and major steps for analysis. What are other feasible alternatives? What are the advantages of adopting this schematic flow over others in this case? How will this affect the results? The authors should provide more details on this.

Response 12: Thank you very much for your comment.

Another feasible alternative to the analysis flow is to use the actual neutron flux measured by the cosmic ray neutron sensors. However, the actual neutron count data are unavailable for three sites (Gorigo, SMEAR II and Conde) hence the synthetic data. Again, with synthetic data, we can assess the actual performance of the filters because the uncertainties introduced to this data are known, unlike observed measurements.

Point 13: p.7 - four error measurements are adopted to evaluate the performance of the filters. What are the other feasible alternatives? What are the advantages of adopting these evaluation metrics over others in this case? How will this affect the results? More details should be furnished.

Response 13: Thank you. Yes, there are other evaluation metrics, however, these statistics (Bias and RMSE) describe the error’s overestimation/underestimation and magnitude simultaneously. A quantitative measure of the degree of dispersion of predictions could be obtained by both RMSE and MBE. The Pearson correlation coefficient is produced via the Pearson product-moment correlation (r). This coefficient efficiently measures the degree of linearity between two variables. Especially with the Pearson correlation, the data satisfy the assumption of normality which studies showed that when the neutron counts exceed 30 cph it approaches a normal distribution (Schron, 2016).

Point 14: p.7 - specific window sizes are adopted to test the performance of three of those filters (MA, MF, and SG). What are the other feasible alternatives? What are the advantages of adopting these window sizes over others in this case? How will this affect the results? More details should be furnished.

Response 14: These filters work based on the principle of the sliding window. Therefore as part of the filters’ algorithm, we specify the window size for filtering the signal. As indicated by the window size significantly affects the signal-to-noise ratio. For this study, we considered window size between 6 – 105 hrs. This range of window sizes enables the study to assess the performance and behaviour of the filters to the window sizes as well as the optimal point of each filter.

Point 15: p.13 - “…However, relatively large window size based on the degree of variance in data may result in the overestimation of the correct neutron flux measurement.…” Some justification should be furnished on this issue.

Response 15: Thank you, the current manuscript has improved. This statement has been changed to more conclusive information in the conclusion section.

Point 16: Some key parameters are not mentioned. The rationale on the choice of the set of parameters should be explained with more details. Have the authors experimented with other sets of values? What are the sensitivities of these parameters on the results?

Response 16: The current manuscript has been improved, especially with the detailed methods and results section. It will be helpful to point out some key parameters the authors should have mentioned if the current manuscript needs to present them.

Point 17: The discussion section in the present form is relatively weak and should be strengthened with more details and justifications

Response 17: We appreciate the thorough reading of the manuscript. The current manuscript has enhanced the discussion of the results and included a new table (Table 2 on p9, line 235) and cited articles to strengthen, justify and improve the understanding of the results.

Point 18: Some assumptions are stated in various sections. More justifications should be provided on these assumptions. Evaluation on how they will affect the results should be made.

Response 18: Thank you for this observation, We have addressed this in our current manuscripts. In fact, your revelation has helped us improve our description, especially in the materials and method section.

Point 19: Moreover, the manuscript could be substantially improved by relying and citing more on recent literature about real-life applications of modeling techniques in soil moisture estimation such as the following. Discussions about result comparison and/or incorporation of those concepts in your works are encouraged:.

Response 18: Thank you very much for the recommended literature.

Point 20: Some inconsistencies and minor errors that needed attention are:

20a: Replace “…Different window size (6 – 105 in steps of 6 hrs) were used…” with “…Different window sizes (6 – 105 in steps of 6 hrs) were used …” in lines 189-190 of p.7

Response 20a: Thank you very much, the sentence has been changed as suggested on lines 234-235 of the current manuscript.

Replace “…We therefore, caution that when applying these filters, identifying window size and interpreting sub-daily soil moisture data from CRNS care should be taken…” with “…We therefore caution that, when applying these filters, identifying window size and interpreting sub-daily soil moisture data from CRNS, care should be taken …” in line 332 of p.14

Response 20b: Thank you very much, the sentence has been changed as suggested on lines 416-418 of the current manuscript.

Point 21: In the conclusion section, the limitations of this study, suggested improvements of this work and future directions should be highlighted.

Response 21: We have revised the conclusions to include some limitations and recommendations in lines 416-421 of the current manuscript.

Round 2

Reviewer 1 Report

I have no further comments.

Author Response

Thank you very much for your review time and suggestions

Reviewer 2 Report

The most significant comments in the previous reviews (including novelty, major difficulties and challenges, their original achievements to overcome them, overlapping with their previous works, etc.) have not been demonstrated satisfactorily. There are still excessive self-citations (six self-citations in total) and the overlapping with their previous works are not clear. No improvement has been made on this issue.

Author Response

Response to Reviewer 2 Comments:

Comment 1: The most significant comments in the previous reviews (including novelty, major difficulties and challenges, their original achievements to overcome them, overlapping with their previous works, etc.) have not been demonstrated satisfactorily. There are still excessive self-citations (six self-citations in total) and the overlapping with their previous works are not clear. No improvement has been made on this issue.

Response: Thank you for your comments.

We modified the abstract to include the novelty, major difficulties and challenges addressed by the manuscript [lines 1-6]:

Uncertainty of the CRNS-derived soil moisture strongly depends on the CRNS count rate subject to Poisson distribution. State-of-the-art CRNS signal processing averages neutron counts over many hours, thereby accounting for soil moisture temporal dynamics at the daily, but not sub-daily time scale. We demonstrate CRNS signal processing methods to improve the temporal resolution of the signal in order to observe sub-daily changes in soil moisture and improve the signal-to-noise ratio overall.

And [lines 13-22]:

Longer window sizes were more beneficial in the case of moderate soil moisture variations during long periods. For MA, MF and SG optimal window size were identified and varied by count rate and climate, i.e. temporal soil moisture dynamics by providing a compromise between monitoring sharp changes and reducing the effects of outliers. The optimal window for these filters and the Kalman Filter always outperformed the standard procedure of simple 24-hour averaging. The Kalman Filter showed the highest robustness in uncertainty reduction at three different locations and maintained relevant sharp changes in the neutron counts without the need to identify the optimal window size. Importantly, standard corrections of CRNS before filtering improved soil moisture accuracy for all filters. We anticipate the improved signal-to-noise ratio to benefit CRNS applications such as detection of rain events at sub-daily resolution, provision of SM at the exact time of a satellite overpass, and irrigation applications”

We have included in the current manuscript paragraph 4 (lines 89 - 124) key achievements, difficulties and challenges, including some overlaps with previous reviews.

We added [lines 89-112]: “Several studies have examined how some environmental factors affect neutron intensity, as seen in the previous paragraph. These early studies focused on reducing uncertainties caused by environmental factors like biomass, water vapour, and incoming neutron intensity on the measured neutron counts. The temporal changes induced by these factors on the CRNS counting rate were studied and corrected. An example of such studies was by [15 ], whose work highlights a correction factor to account for the changes in the cosmic-ray, and high-energy neutron intensity, which affects the epithermal–fast neutron count measured by a CRNS probe. Also, [21 ] developed a correction factor for removing the influence of the temporal changes in atmospheric water vapour content on neutron count. Other studies [15,17,22,24,26 ] contributed significantly to the correction of uncertainties associated with belowground and aboveground biomass. In the case of temporally stable additional hydrogen pools, estimating their contribution and subtracting it from the neutron counts when converting them into volumetric soil moisture contents reduces uncertainty [ 15 ]. According to [ 17 ], more complex corrections are required when hydrogen pools change with time. A systematic uncertainty analysis has been conducted by [27], who quantified how vegetation or soil properties affect the CRNS product. Furthermore, [28 ] proposed an area-sensitivity function based on the number of neutrons emanating from a given radial distance. All these proposed correction functions formed the basis for standard correcting raw neutron counts to remove uncertainty in soil moisture estimates. These studies focus on reducing uncertainties due to some environmental factors. However, additional hydrogen sources must be accounted for, especially if their contributions change significantly over time to reduce error. It is evident that averaging removed high-peaked noise that we attribute to uncertainty in the count rate. Besides, few studies investigate the performance of smoothing algorithms on neutron count uncertainty reduction.”

The summary of this paragraph points out that most of the earlier studies focused on the reduction of uncertainties due to environmental factors such as the water vapour, atmospheric pressure, incoming neutron intensity and biomass together with other unidentified hydrogen pools. As a result, several correction factors were developed and proposed for estimating soil moisture from CRNS data. Furthermore, some of these environmental factors are dynamically changing, which affects the correction process. In addition, since the counting rate of the CRNS detector follows a Poisson statistic, this measured neutron flux can be averaged over time to reduce uncertainty in neutron counting. In order to achieve optimal results, it is critical to choose the smoothing technique or window size correctly.

Nevertheless, no study has yet rigorously examined the effect of these influences on the accuracy of soil moisture measurements with CRNS, and most studies use either 24-hour filters, daily averages, or, in rare cases, two different filtering methods.

This current study employed four different filter algorithms and wide window sizes (6 – 102 hours) to assess the filter performance at different window sizes. The Kalman filter has never been used in a CRNS study to filter neutron counts. Furthermore, these window sizes were assessed for their ability to capture short-term events such as a sudden change in neutron count/soil moisture caused by irrigation, precipitation, or snowmelt.

Also, the six self-cited articles identified are relevant to understanding the background of the CRNS research area. These were key contribution made by these article in the context of factors that influences uncertainties in neutron counts. These articles happen to be relevant to the current study. Besides, the journal did not provide a clear cut on what is considered excessive self-citation. We believe 6 out of 47 references (12.7%) self-citation in the previous manuscript can be considered for any journal publication. The current manuscript self-cited 6 out of 50 (12%) references.